# “You Can’t Work with My People If You Don’t Know How to”: Enhancing Transfer of Care from Hospital to Primary Care for Aboriginal Australians with Chronic Disease

**DOI:** 10.3390/ijerph18147233

**Published:** 2021-07-06

**Authors:** Ilse Blignault, Liz Norsa, Raylene Blackburn, George Bloomfield, Karen Beetson, Bin Jalaludin, Nathan Jones

**Affiliations:** 1Translational Health Research Institute, Western Sydney University, Penrith, NSW 2751, Australia; elizabeth.norsa@health.nsw.gov.au; 2South Western Sydney Local Health District, Liverpool, NSW 1871, Australia; raylene.blackburn@health.nsw.gov.au (R.B.); george.bloomfield@health.nsw.gov.au (G.B.); karen.beetson@health.nsw.gov.au (K.B.); bin.jalaludin@health.nsw.gov.au (B.J.); nathan.jones3@health.nsw.gov.au (N.J.); 3Ingham Institute for Applied Medical Research, UNSW Sydney, Liverpool, NSW 2170, Australia

**Keywords:** indigenous health and wellbeing, holistic health, healthcare innovation, cultural competence and cultural safety, Aboriginal, chronic disease, hospital, discharge, urban, program evaluation

## Abstract

Indigenous Australians experience significantly poorer health compared to other Australians, with chronic disease contributing to two-thirds of the health gap. We report on an evaluation of an innovative model that leverages mainstream and Aboriginal health resources to enable safe, supported transfer of care for Aboriginal adults with chronic conditions leaving hospital. The multisite evaluation was Aboriginal-led and underpinned by the principles of self-determination and equity and Indigenous research protocols. The qualitative study documented processes and captured service user and provider experiences. We found benefits for patients and their families, the hospital and the health system. The new model enhanced the patient journey and trust in the health service and was a source of staff satisfaction. Challenges included staff availability, patient identification and complexity and the broader issue of cultural safety. Critical success factors included strong governance with joint cultural and clinical leadership and enduring relationships and partnerships at the service delivery, organisation and system levels. A holistic model of care, bringing together cultural and clinical expertise and partnering with Indigenous community organisations, can enhance care coordination and safety across the hospital–community interface. It is important to consider context as well as specific program elements in design, implementation and evaluation.

## 1. Introduction

Australia’s Indigenous peoples comprise two distinct and diverse cultural groups: Aboriginal and Torres Strait Islander peoples. Their diversity is exemplified by over 250 language groups across the continent and islands [1]. Aboriginal people are believed to have lived in Australia for over 65,000 years and have the world’s oldest continuing culture. Their deep connections to country, family and community have remained strong despite the impact of colonisation on individual and collective health and wellbeing [1,2].

Indigenous peoples globally face greater social disadvantage and worse health than the general population in their countries [2]. Aboriginal and Torres Strait Islander Australians experience significantly poorer health and wellbeing across the lifespan compared to other Australians [3], with chronic disease contributing to two-thirds of the health gap [4]. They are three times more likely to require hospital admissions for conditions which could have been managed in primary care settings at an earlier stage and are almost twice as likely to have a restrictive long-term condition or a disability [4]. This has been identified as a priority area for better health outcomes [3,4].

Aboriginal and Torres Strait Islander peoples living with chronic disease face numerous barriers in obtaining timely and appropriate health care, managing their chronic conditions and maintaining their health [5,6,7]. Family and community obligations often take precedence [7]. Dispossession, transgenerational trauma, socioeconomic disadvantage and racism and discrimination within mainstream health services affect both access to services and quality of care received [5,6,7,8].

Accumulating evidence that traditional hospital discharge is associated with suboptimal healthcare and avoidable readmissions has resulted in a greater focus on care coordination and transition of care [9,10,11]. Transfer of care initiatives aim to improve care and outcomes following transfer of patients between acute settings (hospital) and primary care (general practice and community) [12]. Culturally sensitive and timely transfer of care planning could help to maintain continuity of care for Aboriginal and Torres Strait Islander patients and improve health outcomes, thereby contributing to reducing the gap between Indigenous and non-Indigenous Australians [12]. There is little published research on hospital-based transfer of care initiatives for Aboriginal patients with chronic disease, particularly in urban settings [13,14]. Studies in Australia and other settler colonial countries, such as the USA and Canada, have largely focussed on primary care and community-based initiatives and rural and remote settings [6,15,16]. In acute settings, the emphasis is often on admission/readmission rates for those with chronic conditions [17,18,19], with limited attention to patient and staff experiences.

New South Wales (NSW), Australia’s most populous state, has 265,685 or 33% of the total national Indigenous population [20]. Building on several NSW Ministry of Health state-wide initiatives to improve coordination and management of care for Aboriginal people with chronic conditions [21,22,23], South Western Sydney Local Health District (SWSLHD) developed a unique Aboriginal Transfer of Care (ATOC) model. Predicated on the assumption that Aboriginal health is everybody’s business [24], the model incorporates a multidisciplinary transfer of care planning process that is person-centred and holistic, in keeping with the Aboriginal concept of health [25]. In contrast to the biomedical model, rather than a single, separate aspect of life, health is viewed as all-encompassing and deeply connected to culture, including country and family, kin and community, and to the past, present and future [25,26]. Recognising peoples’ identities in connection to culture, spirituality, families, communities and country is a core value in a wellbeing model for Aboriginal people with chronic conditions [27].

ATOC was developed and piloted at Campbelltown Hospital in response to the high rate of unplanned readmissions for Aboriginal patients with chronic disease and later adopted at Liverpool Hospital; both hospitals within SWSLHD. This article reports the qualitative component of a multisite, mixed-methods evaluation of the SWSLHD ATOC model. (In this article, Aboriginal and Torres Strait Islander people are mostly referred to as Aboriginal people in recognition that they are the original inhabitants of NSW. The term “transfer of care” is used in preference to “discharge” because it conveys that a patient’s care does not finish when they leave hospital but continues in the community.) The primary purpose of this qualitative study was to explore patient, family and service provider experiences and views and to document and refine the model of care for Aboriginal adults with chronic conditions. The included conditions were cardiovascular disease, diabetes, chronic obstructive pulmonary disease, chronic kidney disease (but not on dialysis) and asthma. Additionally, the study sought to improve both the research capability and research translation capability of the Aboriginal managers and staff involved.

## 2. Materials and Methods

### 2.1. Setting

SWSLHD covers the south western suburbs of metropolitan Sydney. The region combines urban, rural and semi-rural areas and includes the traditional lands of the Dharawal, Gundungurra and Darug nations. Over a million people (roughly 12.5% of the NSW population) live within its catchment, with Aboriginal people comprising 2.1% of the population [28]. Migration and historic settlement patterns have resulted in a diverse Aboriginal community, made up of people from across the state and country. Over a third (36.3%) of the SWSLHD Aboriginal population live in Campbelltown Local Government Area (LGA) and a further 18.7% live in Liverpool LGA [28].

Campbelltown Hospital is a major metropolitan hospital providing a range of services, including maternity, palliative care, respiratory, stroke medicine, surgery and emergency medicine. Management has given considerable thought to making it a place where Aboriginal people (who make up 4.8% of the Campbelltown LGA total population) feel welcome and comfortable. Liverpool Hospital is a principal referral hospital providing referral and district acute services to the Liverpool catchment and higher-level tertiary care for South Western Sydney residents, critical care for rural retrieval catchments, and a supra-regional service for brain injury. Within Liverpool LGA, the Aboriginal population (1.5% of the LGA total population) is overshadowed by the overseas-born population (48% of the LGA total population) [28].

### 2.2. ATOC Model

At its core, ATOC brings together cultural expertise in the form of Aboriginal Liaison Officers (ALOs, whose role is to support Aboriginal patients and their families whilst in hospital) and clinical expertise in the form of Transfer of Care (TOC) nurses from the Demand Management Unit (DMU, tasked with improving patient flow). The ATOC team works across the hospital to supplement transfer of care activities and processes carried out at ward level. A range of hospital clinicians (medical, nursing and allied health) and community-based health and social services contribute to ensuring that the necessary supports are in place for Aboriginal patients to transfer safely back to the community and primary care. The SWSLHD Clinical Nurse Consultant–Aboriginal Chronic Care plays an important consultation and coordination role.

The ATOC model has five key elements:Transfer of care planning by a multidisciplinary team;Ensuring the patient and their family understand the follow-up care plan;Ensuring the patient’s General Practitioner (GP) or Aboriginal Medical Service (AMS) is aware of any follow-up arrangements;Ensuring referrals are organised with community providers;Ensuring the patient has the necessary medications, equipment and written patient summary information prior to transfer of care.

Multidisciplinary transfer of care planning takes place at short meetings or “huddles” which are held at the same time each weekday morning and attended by the ATOC team and other staff from the community-based SWSLHD Aboriginal Chronic Care Program (usually by phone). ATOC team members work together to ensure that the patient and family contribute to and understand the plan, the patient’s primary care provider is aware of the follow-up arrangements and referrals are organised with community and social services (e.g., transport, home care or housing) as needed. A 5-day supply of new medications is provided to cover the period between discharge and GP follow-up. Equipment, such as a hospital bed or shower chair, is organised from an Aboriginal loan pool managed by the Occupational Therapy Department and available at each hospital.

The ATOC model was piloted at Campbelltown Hospital in early 2016 and introduced at Liverpool Hospital later that year. While ATOC has evolved over time, and continues to be refined, the key elements remain unchanged. Figure 1 illustrates the ATOC patient journey (from the community through hospital and back to the community) and the care providers and services involved in ensuring care coordination and safety for Aboriginal patients across hospital-community interface. The ATOC program logic is shown in Appendix A. Both were produced as part of the evaluation.

### 2.3. Study Design and Ethics

The qualitative study adopted a strengths-based/appreciative inquiry approach [29], acknowledging that those on the ground (service users and providers) have essential knowledge about what is working well and why, and what is likely to work in their situation. Conducted in 2018–2019, it was undertaken by university researchers who worked alongside the SWSLHD investigators and the ATOC team at each hospital, using a collaborative, participatory approach: a well-established method of inquiry in Aboriginal health research and evaluation [30,31]. Opportunities for research skills transfer and reflection were embedded throughout.

The overall evaluation was Aboriginal-led from beginning (concept and research design) to end (knowledge translation and dissemination) and underpinned by the principles of self-determination and equity and Aboriginal and Torres Strait Islander research protocols [32,33]. The chief investigator (author NJ) and three of the associate investigators (RB, GB and KB) were Aboriginal. Research partners included SWSLHD, NSW Ministry of Health and Western Sydney University. The SWSLHD Aboriginal Health Directorate works in partnership with local Aboriginal Community Controlled Organisations and communities. Tharawal Aboriginal Medical Service had been involved in establishing the ATOC model through the Campbelltown Hospital Aboriginal Health Committee and supported the study and ethics application, which was approved by the NSW Aboriginal Health and Medical Research Council Ethics Committee. Additional approvals were obtained from the SWSLHD and Western Sydney University Human Research Ethics Committees.

Aboriginal authority over the project was exercised formally in steering committee and working group meetings. Aboriginal involvement at every stage and level ensured that the qualitative study reflected Aboriginal cultural values and that local community protocols were respected and followed. The study prioritised Aboriginal voices and included male and female patients of different ages. Timelines were extended to ensure appropriate community engagement. Preliminary work highlighted the importance of local ownership, hence Campbelltown and Liverpool Hospitals were treated as separate case studies for data collection and the initial analyses.

### 2.4. Data Collection

The qualitative study focussed on the experiences and views of Aboriginal patients and their family/carers, ATOC team members and other hospital staff and community-based service providers from government agencies and non-government organisations.

Primary data were collected through key-informant interviews, which were recorded and professionally transcribed, and observations. In keeping with the purposeful sampling approach [34], potential participants, were identified by the SWSLHD investigators in order to obtain a range of perspectives—positive and negative. The ALOs and TOC nurses were responsible for promoting the research to patients and staff, who were formally recruited and interviewed by the research officer. Observations were made of the ATOC staff at work (e.g., daily team meetings or huddles and case conferences). Documents relating to ATOC’s development and implementation were reviewed (e.g., clinical guidelines and forms, reports and conference presentations).

Semi-structured interview guides for each informant group were developed collaboratively. Patients and their family were asked about their experience being prepared for and leaving hospital and how this affected what happened next (processes and outcomes including unintended outcomes). Present and past ATOC team members and relevant clinicians and managers were asked about their role in delivering or supporting ATOC and for their views on the model of care, its strengths and weaknesses and the barriers and enablers to its full implementation and sustainability. Primary care and community service providers were asked about continuity and coordination of care for Aboriginal adults with chronic conditions before and after ATOC’s implementation. All informant groups were asked for suggestions for improvement.

The research officer adopted a flexible approach to interviewing, negotiating the time and location with participants and accommodating professional and personal priorities and needs, e.g., rescheduling interviews and organising transport for patients as required.

This flexibility built rapport and contributed to participants speaking freely. All participants gave written informed consent and were offered an opportunity to review their transcript.

### 2.5. Analysis

The qualitative analysis involved a combination of inductive and deductive research practices and followed the framework method [35]. Findings were organised according to the *Ngaa-bi-nya* evaluation framework [36], which provides a structure through which to generate insights for the development of culturally relevant, effective, translatable and sustainable programs for Aboriginal people. Building on Stufflebeam’s CIPP model [37], it has four domains: landscape factors (Context), resources (Inputs), ways of working (Processes) and learnings (Products) [36]. Transcripts were coded according to categories generated within the framework using NVivo 12. Coding was done independently by the two non-Aboriginal university researchers; transcripts were compared for consistency and differences resolved through discussion. Interpretation of the interview data was supported by ATOC documents and researcher observations and discussed in the qualitative working group (including IB, LN, RB, GB, KB and NJ). Analysis of early interviews suggested additional questions for later interviews and other potential informants.

Emerging findings and insights were discussed at meetings of the qualitative working group and presented to the two ATOC teams for verification and to support continuous improvement. In the latter part of the project, the Liverpool ATOC team undertook an 8-week trial of a revised ATOC Huddle Form. In this participatory action research component, the team decided the length of the trial and what they wanted to learn from it and provided feedback as the trial progressed.

Initial within-case analyses produced a description of the ATOC model at each facility, including its current operation and implementation history, which was reviewed with each ATOC team. Subsequent cross-case analysis provided a basis for overarching insights and lessons. Rigour was maintained through double coding, triangulation of data sources and methods and member checking with the ATOC teams. Regular exchanges between researchers and SWSLHD investigators ensured Aboriginal perspectives were preserved and prioritised during analysis and interpretation.

## 3. Results

### 3.1. Participants

Forty-nine people were interviewed—see Table 1. The eight patients included four men and four women; two were in their 30 s and the rest were over 55 years old. The two female carers were family members (daughter and wife). Interviews were conducted from a couple of weeks to a few months after the patient left hospital. Two patient interviews were conducted in the Aboriginal patient and family room Campbelltown Hospital (the Uncle Ivan room), one at a shopping centre, one at a restaurant and four at their local library.

Other hospital staff included managers, ward nurses, allied health professionals (social work, occupational therapy, physiotherapy), doctors (consultant, registrar, GP) and pharmacists. Community-based service providers included people affiliated with SWSLHD community health programs, Aboriginal Medical Services, South Western Sydney Primary Health Network and the NSW Department of Housing.

Findings from the cross-case analysis are presented in Section 3.2, Section 3.3, Section 3.4 and Section 3.5 below, with quotes labelled according to participant number, site and informant group/role. In documenting the model of care and explaining what works well, we are conscious of the need to consider local context and influences; these are addressed in Section 3.6.

### 3.2. Resourcing and Ways of Working

According to senior managers interviewed, one of the ATOC model’s attractions was that it was “resource neutral” from a hospital perspective. The model relies on bringing together existing staff (ALOs and TOC nurses), using current technology and working in new ways. The cost of establishing an Aboriginal equipment loan pool was borne by SWSLHD.

Each of ATOC’s five elements contributes to effective transfer of care and a positive patient journey.

In Element 1, multidisciplinary transfer of care planning, the ALOs bring cultural expertise and community knowledge and connections, together with an understanding of the patient’s personal needs, preferences and family circumstances, and the TOC/DMU nurses bring clinical expertise and resources. The following quotations, from an ALO and nurse respectively, are illustrative.


*We would look at the home situation, we would look at—if there was a carer, were they managing? We would look at––did the carer need more support? What was going on at home and was there a carer?*
 (P48, Liverpool Hospital ATOC).

*The ALO is not a nurse and doesn’t have the clinical knowledge and expertise that senior nurses have. I was able to inform and guide the ALO regarding different medical conditions and the expected outcome, recovery rate and length of stay for the ATOC patients*. (P31, Campbelltown Hospital ATOC).

The huddles (short meetings) provide the main platform for information exchange and joint transfer of care planning. They work best when they are tightly structured, participants come prepared and it is clear for each patient who is responsible for follow-up action and reporting back the following day.


*Having that meeting was the first big step … helping identify the patients, their care needs … facilitating discharging them and making their transition home as best possible.*
 (P1, Liverpool Hospital manager).


*Having a quick 15-meeting to go “Right, these are the patients in hospital. I’ve seen this one. I’ve seen that one … This one’s got these problems. This one’s well set up at home. This one just needs transport sorted to get home”. That type of thing.*
 (P31, Campbelltown Hospital ATOC).

Element 2 involves ensuring that the patient and their family understand the follow-up care plan. For some patients and family/carers lack of communication was “*the biggest problem”.* The ALO’s role in helping people feel safe by providing reassurance and information was valued by everyone. One patient who came from a regional area commented:


*[The ALO] talks to you on a blackfella level, the way they should, especially in the city … He tells the ins and outs of everything, explained everything.*
 (P45, Liverpool Hospital patient).

Among service providers, in addition to supporting smooth transfer of care, timely communication was seen as critical to avoiding discharge against medical advice and preventing unplanned readmissions and Emergency Department (ED) presentations within 28 days (all health service metrics).


*So, I think communication is the key because I think once it gets to the point of no return, people often will discharge themselves against medical advice. When they don’t really need, we should never have got to that point … And so then they don’t get the optimal care they need, they don’t get the follow up they need. Often they leave without medications so then you’re putting them at higher risks of adverse events.*
 (P28, Campbelltown Hospital ATOC).

Many Aboriginal people admitted to Campbelltown and Liverpool Hospitals are registered at an Aboriginal Medical Service; others have a regular GP. Any patient without a primary care provider is connected with one. Those with chronic conditions are offered referral to the SWSLHD Aboriginal Chronic Care Program through which they can access care coordination, specialist services and supplementary services.


*[Aboriginal chronic care] works best when we look at things holistically and when we’re looking at absolutely everything—the psychosocial issues, the social determinants, where people are storing their medication, food—absolutely everything … ATOC really brings those conversations to light and if those conversations come to light very early, means we’re looking at things holistically.*
 (P22, Community-based service manager).

Referrals for home care services (e.g., help with housework, meals or shopping) and social housing assistance are common. Several informants referred to accommodation as no longer being suitable or contributing to chronic ill health.

*Aboriginal patients’ housing and their home environment comes up a lot at our meetings … It’s usually a medical condition, failing heart or lung disease, and they can’t get up and down the steps or its too far to walk or it’s not close to the shops anymore*. (P30, Campbelltown Hospital ATOC).


*And at home it’s so damp … I’ve got all mould in the house. They’ve come and inspected my house for the last five to seven years and the moulds been there and nothing. ‘Oh, yeah, we’ll fix it up. We’ll fix it up.’ Nothing. And this is why I keep going in hospital, because of the mould too. I can’t breathe.*
 (P13, Liverpool Hospital patient).

Finally, a TOC/DMU nurse is responsible for reviewing each patient to ensure they have the necessary medications, equipment and written patient summary information before they leave hospital. On top of the usual three-day supply, a further two-day supply of new medication has been negotiated for all SWSLHD Aboriginal patients. Having equipment, such as a hospital bed or shower chair, organised and set up at home prior to leaving hospital reduces stress for the patient and their family and lessens the workload for primary care providers.


*A hospital bed is one of the biggest things that needs to be organised before they go home to make them comfortable in their bed. Equipment needs to be organised … so that [there is] less stress for the family when they go home”.*
 (P18, Community-based service provider).

### 3.3. Outcomes and Impact

The pilot study at Campbelltown Hospital showed an immediate effect, recording a steady decrease in Aboriginal patient unplanned readmissions and ED presentations over four months, as well as an increase in Aboriginal patient identification in ED. After 2–3 years of operation, this qualitative study found better coordination of care and improved transfer of care experience. ATOC patients felt safe and supported going home and working as a team towards a common goal was a source of satisfaction for ATOC staff. Benefits reported at the organisational and system levels included increased trust in the health service and stronger partnerships between services.

For many Aboriginal people living with chronic conditions, especially those with frequent hospital admissions, a sense of safety may be elusive. Two male patients admitted after a heart attack, both in their mid-30 s, were badly shaken. Having the ATOC team to provide support and the ALOs to explain what is happening and what will happen, including follow-up arrangements, is reassuring for patients and their families.

*I had a lot of support there which was good because that’s what you really need. You’re in foreign place. You’re scared. You don’t know what’s gonna happen*. (P13, Liverpool Hospital patient).


*It’s just a bit scary coming out of hospital trying to, after what’s happened to you … Because most of the time I’ve been in there it’s life-threatening, it gets scary. It gets scary in there and it’s scary out here.*
 (P43, Campbelltown Hospital patient).


*It just makes me feel at ease, really at ease … I’ve got someone there to help me. I’m not on my own with the system.*
 (P41, Campbelltown Hospital patient).

Increased communication leads to increased awareness; this is particularly important for TOC/DMU nurses who may join ATOC with little knowledge of Aboriginal health and history or the local Aboriginal community. Through dialogue with patients and their family, they come to understand their issues and risks. For Aboriginal patients, having a positive experience during and when leaving hospital can help build trust and relationships and over time lead to improved engagement with the health service and reduced self-discharge.


*We had one Elder that came in that it took a while for him to get into hospital. However, once he was here and we did speak to him and we did support his progression here in the hospital. Once he was discharged, he went home happy and we got the feedback from [the ALO] that he has been trying to encourage other Aboriginals that he knows that are very sick to come in to hospital because we will help them and that we are providing a good service and acknowledging their culture and supporting their culture.*
 (P29, Campbelltown Hospital ATOC).

ATOC staff feel satisfied and are motivated to continue working in this demanding area when they feel they are part of a team and know, through feedback about individuals and service reports, that their efforts are “helping to close the gap”. Acknowledging that the influences on health are multifactorial and that change often takes time, staff recounted numerous examples of positive outcomes resulting from patient-centred, multidisciplinary care planning and coordination across the hospital–community interface.


*I think it’s very necessary and it helps across the board of the hospital’s operations in terms of patient flow, patient satisfaction, and in a way, staff satisfaction. I always feel really happy when we have a successful ATOC case that doesn’t represent to hospital and it’s like ‘Wow! We’ve done everything that we can for you and you’re managing very well at home. You’re not engaging all of these high-risk behaviours and all that stuff’.*
 (P49, Liverpool Hospital ATOC).

Positive experience with ATOC has led to the development of new Aboriginal health collaborations and strengthened existing partnerships and systems. Shared-care meetings for highly complex patients were described by participants as hugely beneficial. The following comment from a Housing officer indicates how well they see the model working:


*[Department of] Housing are getting phone calls earlier from staff in the hospitals about saying ‘We have someone came in last night that looks like they’re going to be in here for the week. They’re homeless. Can we get some information about their housing?’ So that’s been really beneficial and it’s been vice versa when I know that I’ve had, especially homeless clients, because they’re so transient when I know that they got into hospital, I can call that hospital and say ‘These are the issues. Can we try to get a care coordination while they’re there, before they leave hospital as well?’*
 (P24, Community-based service provider).

### 3.4. Challenges

In a busy hospital with the inevitable pressure on beds and patient flow, ensuring that all Aboriginal patients with chronic conditions leave with the necessary supports in place to be safe and well in the community is not easy. Key challenges include staff availability, patient identification and complexity and the broader issue of cultural safety across the hospital.

ATOC typically operates from 9.00 a.m. to 5.00 p.m. Mondays to Fridays, which affects service delivery and limits patient access. Outside the huddles, ATOC relies on routine hospital information systems and communication tools. Incomplete identification of Aboriginal patients, and for ATOC those with chronic condition, is an ongoing problem. The transition from paper notes, where Indigenous status was identified on the front page, to an electronic medical record requires that clinicians routinely check the relevant fields. Once a patient is identified, ATOC is intended as a “short and sharp” intervention. The model does not work well—and was not intended for—complex patients who require additional time and effort; the first step is to identify and prioritise their needs and then to ensure they are all addressed.

Poor treatment of Aboriginal patients was something that both service users and providers commented on:


*I don’t think they treat the Aboriginal people the way they should be treating them and that’s why a lot of Aboriginal people will not go to hospital. They will not go to a doctor and they will not go to a hospital because the way they’re treated.*
 (P13, Liverpool Hospital patient).


*Many patients have historically not had positive experiences with hospitals. [We try] to ensure ATOC patients feel comfortable and safe; this encourages the patient to remain in hospital until they are well enough to go home and be compliant with medications and follow-up.*
 (P31, Campbelltown Hospital ATOC).

Building rapport with Aboriginal people who have not been well-served by government health services in the past, and gaining their trust, takes time and effort. Lack of cultural safety and misunderstanding of the concept of “equity” as opposed to “equality”, with staff seeing more value in treating everyone the same rather than “respecting the difference”, is a major challenge. Several informants recounted instances of discharge against medical advice when patients felt their needs were being ignored.

### 3.5. Critical Success Factors

From the outset, ATOC has benefited enormously from good will and shared purpose combined with a pragmatic, problem-solving approach. Critical success factors include strong governance structures with joint cultural and clinical leadership, plus strong and enduring relationships between individuals and teams at the service delivery, organisation and system levels.

ATOC is one of several Aboriginal health initiatives in SWSLHD and sits under a broad governance framework. At the District level there has been joint leadership from the Aboriginal Health Directorate and Nursing. Each hospital has its own Aboriginal Health Committee and there has been ongoing support from the executive. A shared understanding of ATOC’s rationale and process is fundamental. Senior managers commented on commitment from management and staff:


*I think the fact that there’s a general commitment within this agency that Aboriginal health is a priority has helped us push a lot of this along.*
 (P17, SWSLHD manager).

*If you develop the systems and some good governance and develop what’s important, you can then start to develop your culture around [that] … It’s the culture and your staff, your staff commitment to those processes that will keep them going*. (P33, Campbelltown Hospital manager).

On the ground, safe transfer of care is enabled by a skilled and culturally-competent workforce and a collaborative approach. Within the core ATOC team, ALOs bring cultural expertise and TOC/DMU nurses bring clinical expertise. Two ALOs, female and male, are desirable to uphold cultural gender protocols, provide peer support and share the workload. Cultural training is essential for non-Aboriginal ATOC staff, broadening their focus to include the social determinants of health and a cultural perspective.


*You can’t work with my people if you don’t know how to.*
 (P14, Liverpool Hospital patient).


*We know how to communicate ... We now know how to address sensitive issues. We know how to provide the proper support … When a patient or a person feels supported, they then cooperate with the rest of the planning of their stay here. So I think the fact that we’re now, one, culturally aware; two, have [the ALO] and [the ALO] has us to support [them] with the medical side of things provides effective patient care.*
 (P30, Campbelltown Hospital ATOC).


*Having this ATOC team for our patients is just another stepping stone for our people.*
 (P26, Campbelltown Hospital ATOC).

Nevertheless, establishing a small ATOC team is not sufficient. ATOC’s success depends on active participation by other hospital staff involved in the patient’s care and managerial support, as well as effective interagency partnerships. Partnerships with the Aboriginal Medical Services in Campbelltown and Liverpool and the NSW Department of Housing, and recent expansion of the community-based Aboriginal Chronic Care Program, have been particularly important. As illustrated above, access to suitable housing is a critical part of safely transferring patients out of hospital and keeping them healthy at home. Through partnerships and formal and informal links with other services, staff are able to advocate on behalf of patients and their families. Having the ability to escalate an issue to a higher level when necessary and having wide networks makes a big difference, especially where resources are limited—“You have to utilise everything that’s out there”.

### 3.6. Contextual Similarities and Differences

At both study hospitals, there was a stated commitment to culturally-responsive care and recognition of the need for an Aboriginal-specific transfer of care protocol and the crucial role of the ALOs. Identification of Aboriginal patients, adults with a chronic condition in particular, was an ongoing challenge due to incomplete or inaccurate records. Delays in recruitment left the second ALO positions vacant for lengthy periods. Making the intervention part of routine care for the target population was more challenging at Liverpool Hospital (a principal referral hospital), where there was a higher ATOC nursing staff turnover, greater cultural and linguistic diversity among patients and proportionally fewer Aboriginal patients. The hospital environment was less welcoming for Aboriginal people. Further, the sense of ownership, which was very apparent at Campbelltown Hospital where the model was developed and piloted, was lacking. Additional support and resources are needed to implement and sustain the model in such contexts.

## 4. Discussion

Mainstream health services struggle to meet the needs of Aboriginal and Torres Strait Islander patients with chronic conditions [5,13,22,27]. The SWSLHD ATOC model emerged as a local response to a widespread problem. This holistic model of care harnesses cultural and clinical expertise and available resources to enable safe, supported transfer of care for Aboriginal adults with chronic conditions leaving hospital. The qualitative study found the ATOC model was welcomed by Aboriginal patients and their families and by community-based health and social service providers. The new model enhanced the patient journey, improved the patient and family experience and trust in the health service and was a source of staff satisfaction. From a hospital perspective, the cost of ATOC implementation was minimal. Originally designed for patients with chronic conditions, the model has been extended to cancer and palliative care. Service providers and managers saw considerable merit in extending it further (e.g., to mental health and maternity).

The findings on challenges and critical success factors are consistent with those reported in an earlier review of Aboriginal transfer of care initiatives in Australia [12]. In that work, commonly identified barriers included system complexity; lack of clear, established and well-understood pathways; and lack of an appropriately-trained and coordinated workforce with clear roles and responsibilities. Local referral pathways were considered an enabler and the roles of Aboriginal Health Workers were consistently highlighted [12]. In this study, key informants also raised the broader issue of cultural safety and its impact on Aboriginal people’s experiences of hospital care.

Around the world, different and shared experiences of racism, including systemic racism, continue to adversely impact the health of Indigenous peoples [38,39,40]. Long-term structural inequities continue to play out at the point of care, where negative experiences inform Aboriginal people’s level of trust of the health systems [41]. Cultural security has been described as a key element of effective services for Indigenous peoples globally and “an ongoing journey” [42]. Among health and social service providers working with Aboriginal families in Western Australia, only 73% considered their service to be culturally secure. Employing Aboriginal staff and better cultural awareness training for non-Aboriginal staff were highlighted as ways to improve cultural security [42].

Cultural competency, safety and security are required at both individual health practitioner and organisational levels to achieve equitable healthcare delivery [43]. (In Australia, the terms cultural awareness, competency, safety and security are often used interchangeably, although they have different emphasis and implications [42]. We have used the terms employed by our informants and in the literature.) Cultural training delivered across the organisation is an important part of this, as is the inclusion of Aboriginal health workers in healthcare teams [8,13,44,45]. *Respecting the Difference* is the Aboriginal Cultural Training Framework used by the NSW Ministry of Health and SWSLHD [46,47]. The unique contribution of Aboriginal health workforce—bringing connections to community and cultural and spiritual knowledge, providing culturally-appropriate healthcare and influencing the health services within which they work—is well documented [48,49]. Research in hospitals highlights the indispensable role of ALOs in improving communication between patients and clinicians and continuity of care and in reducing discharge against medical advice, for Aboriginal adults admitted to cardiac and mental health wards [13,14,50,51]. In the ATOC model, ALOs and TOC nurses work together and with others to address the cultural, clinical and psychosocial aspects of a patient’s transfer of care.

Beyond the ATOC model itself, our findings suggest that there is considerable merit in the SWSLHD Aboriginal governance framework which encompasses all district hospitals and other Aboriginal health initiatives, such as the SWSLHD Aboriginal Chronic Care Program and partnership arrangements with Aboriginal Community Controlled Health Organisations, that have enabled the model’s development and growth. ATOC aligns well with *He Pikinga Waiora*, a theoretical framework that argues that implementation science for Maori and other Indigenous communities should be grounded in Indigenous knowledge, participatory approaches and systems thinking [52,53]. The *He Pikinga Waiora* framework incorporates four elements, all of which are reflected in the ATOC model implementation and evaluation: culture-centred approach, community engagement, systems thinking and integrated knowledge translation [52].

The qualitative study with its collaborative, participatory approach produced immediate benefits and strengthened both the research capability and research translation capability of the Aboriginal managers and staff involved. Steering committee and working group meetings and data collection activities provided many opportunities for mutual/two-way learning [54]. Data collection created space for ATOC team members to reflect on their day-to-day practice, contributing to better understanding of the model and their respective roles. At Liverpool Hospital, the ATOC model was refined, with the team testing a revised version of their ATOC Huddle Form. The short trial resulted in improved information exchange and changed the dynamic of the huddle. At Campbelltown Hospital, the research confirmed existing practices and enhanced the team’s profile. At both sites, the ALO role was validated and their voice strengthened. Aboriginal patients and carers were also given a voice. Connections with community-based health and social services were strengthened. The SWSLHD investigators learned about different research methods and their strengths and limitations and developed their public-speaking skills through presenting at conferences. Aboriginal Health managers gained experience in leading a large evaluation, in partnership with university researchers. This included “real time analysis” and using interim findings to improve service delivery and systems. ALOs gained a “better appreciation of research” and enjoyed “sharing what we’re doing with other services”.

Evaluation outputs included two educational posters and a toolkit to strengthen implementation and support transfer to other settings wishing to adopt the model. The posters were developed to explain the model and emphasise the importance of teamwork, support structures and partnerships [55,56]. Both incorporate Aboriginal design and artwork by a local Aboriginal artist. One poster includes a ‘cultural yarn’ (developed by KB) telling the ATOC story and highlighting the value of knowledge sharing across different Aboriginal lands—*The Lyre Bird and the Goanna*. The posters have been embraced by both teams and are used to inform others, within and outside SWSLHD, about ATOC. The *Aboriginal Transfer of Care (ATOC) Model Toolkit* contains an orientation package, roles and responsibilities, five case studies and an ATOC Huddle Form which provides a structure for the short team meetings.

## 5. Strengths and Limitations

The strengths of this qualitative study lie in the use of multiple sites, albeit within a single Local Health District in the Sydney metropolitan area, plus prolonged engagement, triangulation of multiple data sources and modes of data collection and member checking. The purposeful sample approach resulted in a range of viewpoints and experiences. Aboriginal perspectives were preserved and prioritised during analysis and interpretation. The study provided insights into process and service user and provider experiences, as well as outcomes. It was able to take into account the ATOC model’s evolution over the study period and the differences in implementation at Campbelltown and Liverpool Hospitals. While one must be cautious about transferability to hospitals in other Australian settings and other settler colonial countries, the findings are illustrative of what can be achieved in this area and provide valuable learnings. Given the complexity of health and social issues experienced by the ATOC patient cohort and the multiplicity of agencies often involved in their care, outcomes cannot be attributed to ATOC alone.

## 6. Conclusions

This collaborative research project examined the impact of an innovative model of care which leverages mainstream and Aboriginal health resources to improve transfer of care from hospital to primary care for Aboriginal adults with chronic conditions. The multisite evaluation provides evidence that a holistic model of care based on bringing together cultural and clinical expertise and partnering with Aboriginal community organisations can enhance care coordination and safety for Aboriginal patients across the hospital–community interface. It is important to consider context, including organisational and environmental factors, as well as specific program elements in design, implementation and evaluation. Further research is needed into the contribution of transfer of care initiatives to achieving equitable health care and improved health outcomes for Aboriginal and Torres Strait Islander Australians.

## Figures and Tables

**Figure 1 ijerph-18-07233-f001:**
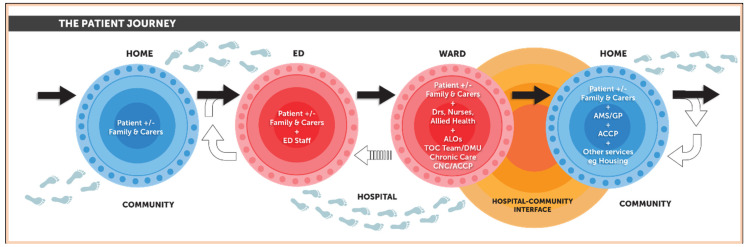
ATOC patient journey. ACCP, Aboriginal Chronic Care Program; ALOs, Aboriginal Liaison Officers; AMS, Aboriginal Medical Service; CNC, Clinical Nurse Consultant; DMU, Demand Management Unit; Drs, Doctors; ED, Emergency Department; GP, General Practitioner; TOC, Transfer of Care.

**Table 1 ijerph-18-07233-t001:** Participants by informant group and site.

Informant Group	Campbelltown Hospital	Liverpool Hospital	Other	Total
ATOC team members	6	4	0	10
Other hospital staff	10	10	0	20
Community-based service providers	0	0	9	9
Patients and family/carers	5	5	0	10
Total	21	19	9	49

## Data Availability

The qualitative data are not publicly available as they contain information that could potentially re-identify individuals. The *Aboriginal Transfer of Care (ATOC) Model Toolkit* is available from Nathan Jones (nathan.jones3@health.nsw.gov.au).

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
