# Peer review of "“You Can’t Work with My People If You Don’t Know How to”: Enhancing Transfer of Care from Hospital to Primary Care for Aboriginal Australians with Chronic Disease"

_ijerph, 2021, doi:10.3390/ijerph18147233_

Round 1

Reviewer 1 Report

Thank you for the opportunity to review this paper on a very important topic. All too often patients experience discharge rather than supported transfer of care, leading to preventable readmission; this has particular implications for Aboriginal and Torres Strait Islander peoples in Australia who experience additional complexities with chronic conditions due to colonisation impacts and racism.

I have some questions for clarification and to help inform future studies.

INTRODUCTION:

Transfer of care versus discharge

  • It may be useful to include a sentence that describes how tertiary hospitals have discharged patients rather than transferred care, generally, and that this is an area many hospitals are seeking to improve.

Aboriginal and Torres Strait Islander population

You mention on pages 2 and 3 the proportion of Aboriginal people in the population. On page 2, are the Aboriginal and Torres Strait Islander peoples in NSW 33% of the total Australian Indigenous population?

South West Sydney – Aboriginal people make up 2.1% of the population there. When you refer to ‘district’ is this South West Sydney?

Within this area there are different Local Government Areas

  • Campbelltown –36.3% of all Indigenous peoples living in SWS district live in this LGA
  • Liverpool – 4.8% of all Indigenous peoples living in SWS district live in this LGA. There is also a higher proportion of international born population in this LGA
  • Do many people identify as Torres Strait Islander in SWS or is it mainly Aboriginal peoples from across NSW?

  • Discharge medications – extended from 3 days supply to 5 days. Are patients able to see their GP within 5 days. In many places across Australia there is a waiting list up to 2 weeks to see a GP. Is there also a process to ensure an appointment is booked in time?
  • The Aboriginal loan pool (p3), is this based at each hospital?
  • Figure 1 – the patient journey is a very clear diagram.
  • However, I found reading through that I struggled to keep track of the acronyms of each team and understand each of the people involved in ATOC and where they are placed. A table could accompany Figure 1 with 2 columns, Hospital/ward and Home with each of the teams and roles listed below.  The acronym and full text could also be in this table for a quick reference guide.

DATA COLLECTION AND ANALYSIS:

  • I note that the data was collected and initially analysed by non-Indigenous members of the team, and then discussed with Indigenous team members. We have been thinking deeply about this in our own projects - about how questions are framed, answered and interpreted according to our own world views and cultural backgrounds, and I am interested to know if this came up as a consideration or issue within this study. I note that there were regular exchanges to ensure Aboriginal perspectives were preserved and prioritised. My question is perhaps more about the verbal and non-verbal cues exchanged during interviews and the interpretation of these. You have provided a discussion of research rigour, this is a question related to cultural interpretation of data, which may be covered in your wider research team discussions of emerging themes?
  • Were there options for choice of cultural background of interviewer, particularly for ALOs and patients, and was this a consideration? Did gender of interviewer impact on participants, and were choices available/offered? Or did this not arise as an issue or consideration?
  • The “research officer” (page 5) is that the person interviewing, and were there two people or only one interviewing?
  • Where did the interviews of patient participants take place? In the hospital or at home, and was there a particular period of time after transfer of care (this is a question to help inform future studies – the time frame is something we have discussed at length, and with ethics).
  • Patient journeys - I note that thematic analysis was undertaken. Was there also any specific journey mapping of individual journeys?

Participants

  • There were 8 patient participants - (most over 55) – page 5. Then later in the quotes two of the men are in their 30’s having had heart attacks – page 7 - which is very young. Perhaps start with more detailed or comprehensive description of the patient participants. The two carers, are they family members?

  • I note that there are nursing, ALO and management participants, but there do not appear to be any medical participants. Is this because they were not directly involved in the project, and transfer of care arrangements?

RESULTS

  • Table 1 - The project obviously focuses on the hospital side of the informants but I am also interested to know who made up the PHC and other support service informant group. Rather than ‘other’ for everyone out of the hospital sector, is it possible to break down the community based service providers into PHC and support services. Is it possible to divide further into GP/AMS or PHC/AMS, or were these mixed services where this is difficult to identify? My reason for asking is for future studies and programs to consider who may be involved.

DISCUSSION

  • You mention that the model has been extended to cancer and palliative care. Has this occurred after this study – ie is this study reporting on chronic conditions, but there may be future studies reporting on cancer, palliative care and possibly mental health and maternity journeys?

  • Cultural models. There are a range of cultural approaches discussed – cultural awareness, competency, security and safety. While connected, these all have quite different underpinning philosophies and emphasis and impact differently in mainstream hospitals – at individual health practitioner and organisational levels. For example cultural awareness sand cultural safety training for staff are quite different in content, intention and impact, as discussed by Downing, Kowal Paradies  (2010, 2011), Kerrigan et al 2020, etc

  • Is there any linking between the priorities of this study – with the National Safety and Quality Health Service (NSQHS) Standards – which prioritise Partnering with Community (Standard 2 and Action 2.13), cultural awareness and cultural competency (Action 1.21), Comprehensive care (Standard 5) and identification (Action 5.8). I am interested as this has been a significant point of interest for the hospitals we are working with in another jurisdiction in Australia.
  • On page 12 paragraph 1 “The purposefully invited volunteer sample “– what does this refer to exactly?

Reviewer 2 Report

Please refer to the attached word document.
